# Dropping Fire Retardants by Helicopter and Its Application to Wildfire Prevention near Electrical Transmission Lines

Jiazheng Lu [1], Tejun Zhou [1,2,*], Chuanping Wu [1,3] and Yangyi Ou [1]

[1] State Key Laboratory of Disaster Prevention and Mitigation of Power Transmission and Transformation Equipment, Changsha 410007, China
[2] College of Electrical and Information Engineering, Changsha University of Science and Technology, Changsha 410007, China
[3] Hunan Disaster Prevention Technology Co., Ltd., Changsha 410007, China
* Correspondence: zhoutejun1988@126.com

**Abstract:** Dropping fire retardants by helicopter can effectively reduce the intensity of wildfires. This study proposes a test plan for spraying different fire retardants from a helicopter bucket fire extinguisher. In this study, pure water, 10% Class AB flame retardant, 0.3% gel flame retardant, 10% Class A flame retardant, and 10% Class A flame retardant + 0.6% guar gum were each added to the bucket fire extinguishing device and sprayed on 4-layer, 6-layer, and 12-layer wood cribs. The radiation intensity, mass loss, and temperature were used as indicators to compare the burning intensity of the fire field and the difference in fire field combustion intensity after the wood cribs were ignited 1 h after natural air drying. The results showed that flame retardancy could be ranked from high to low as follows: 10% Class A flame retardant + 0.6% guar gum > gel flame retardant > 10% Class A flame retardant > Class AB flame retardant > pure water. During the long-term high temperature and drought period in Hunan Province, China, from August to September 2022, a field application showed that dropping fire retardants by helicopter effectively reduced the intensity of wildfires and avoided transmission line trips due to the wildfire, which reduced the number of ground personnel required when fighting large-scale forest fires.

**Keywords:** wildfire near transmission line; helicopter; flame retardant

## 1. Introduction

The frequency and intensity of global extreme disasters have increased [1,2]. Furthermore, the high incidence of forest fires poses a major threat to the natural resources provided by forests and grasslands and to human health. Wildfire disasters have also caused major damage to public facilities, such as power grids and communications [3–6]. For example, in California and other regions where large-scale wildfires occurred every year from 2017 to 2020, the high temperature and smoke from wildfire disasters near transmission lines led to a decline in the insulation performance of air gaps, which caused a large number of transmission lines to trip, resulting in large-scale power outages. On average, there are more than 70,000 wildfires in China every year and they have been one of the major causes of power transmission failure over the past 10 years. In 2013, the 1000 kV Changnan Line tripped several times due to wildfires and the power outage lasted 60 h in total. In August 2013, extreme dry weather caused 10 wildfire trips in the Hunan Power Grid area over 13 days from 5 August to 17 August [7]. In August 2022, countries across the globe experienced extreme drought with surging electricity loads. The wildfire in Banan District, Chongqing, China, caused the 500 kV Luonan Line 1 and Line 2 of the outgoing line from Banan Power Plant to trip and shut down for 40 h, which affected power grid operations. The United States, China, and other countries have carried out research on prevention and control technologies, such as prediction, monitoring, and extinguishing wildfires near power transmission lines [8–12]. Kal'avský, Peter, et al. [8] investigated the

effect of water mist on the development of a long-gap discharge and provided guidance for the safe application of water mists to extinguish wildfires near high-voltage transmission lines, and Trakas proposed stochastic mixed integer programming with quadratic constraints to increase the resiliency of a distribution system threatened by wildfire [11]. Firefighting is the most effective way to prevent transmission lines from tripping due to wildfires. However, fires around transmission lines can be difficult to control due to high mountains, precipitous paths, and extremely high emergency time limits. It takes a long time to extinguish the fire using ground fire trucks and backpack firefighting equipment, which means that the fire spread can become larger. Helicopters are important items of equipment that can be used to extinguish forest fires because they can overcome terrain restrictions. In addition, firefighters have no direct contact with the fire scene and are less exposed to danger. There are three advantages to using helicopters and fire retardants to extinguish wildfires near transmission lines: (1) During large-scale wildfires, the fire safety risk to ground personnel is relatively high. On 30 March 2019 and 2020, wildfires occurred in Liangshan and Sichuan, China, resulting in the deaths of 30 firefighters and 19 firefighters, respectively. Dropping fire retardant water agents by helicopter can reduce the intensity of wildfires, which is conducive to ground fire safety and improves the safety of ground firefighters. (2) The number of helicopters available for firefighting is limited when dozens of wildfire disasters occur near the transmission line, which means that it is impossible to quickly put out on-site wildfires in a short period of time. Spraying flame retardants onto the combustibles between the fire site and the line can form an isolation zone to prevent the wildfire from spreading to the transmission line and reduce the risk of large-scale power outages caused by the simultaneous or sequential shutdown of multiple lines due to wildfires. (3) Existing studies have shown that the main factors causing line tripping by wildfires are the high temperature of the flame, flame ionization, and distortion to the electric field caused by the smoke and dust, which are likely to cause streamer discharge [13,14]. Spraying flame retardants on the surface of combustibles in corridors along both sides of the transmission lines can reduce the intensity of wildfires, reduce ionization, and further reduce tripping by wildfires. Therefore, it is important to study helicopter spraying of flame retardants to reduce the intensity of wildfires.

Helicopters and retardants could be used to pretreat areas at high risk of fire ignitions to make them fire resistant [15,16]. There have been many international studies on the formulation of flame-retardant water agents, their flame-retardant mechanism [17–20], and their impact on the environment [21–23]. However, few studies have quantitatively analyzed and evaluated the flame retardant effect on forest fires. Some previous studies on the subject include the following: Ref. [24] measured the solid density and the gas temperature to determine the effect of the chemical agent Phos-Chek, a commercial flame retardant widely used by firefighters in North America, on pyrolysis and flame-spread on untreated wood samples and wood samples treated with Phos-Chek; Ref. [25] describes the application scenarios for long-term retardants, fire suppressant foam, wet water, and water enhancer (gel); and Ref. [15] describes the effect of a sprayable, environmentally benign, viscoelastic fluid comprising biopolymers and colloidal silica that was sprayed using a backpack onto a layer of grass taped to a wood slab. The nozzle was placed ~30 cm away from the grass and sprayed in bursts. Temperature and normalized area burned over time were used to demonstrate the amount needed to prevent the spread of fire. However, the existing literature does not directly compare the fire intensity changes after spraying with flame retardants and the tests were conducted by spraying flame retardants onto the ground. The distribution of flame retardants sprayed on the ground differs greatly from those sprayed by helicopters. Therefore, it is not possible to apply the existing conclusions to the protection of transmission lines on site using a helicopter.

Thus, the main purpose of this study was to undertake in-flame retarding tests using helicopter bucket spraying of an extinguishing agent. The aims were to evaluate how well different fire retardants at different potential fuel concentrations decreased fire intensity and slowed the advance of a fire by measuring the temperature, heat flux density, and mass

loss of forest fuel and by analyzing the fire retarding mechanism in combination with the chemical components of the extinguishing agent.

This paper consists of the following sections: Part I provides an overview; Part II describes the in-flame retarding test program using helicopters; Part III discusses the in-flame retarding test data; Part IV provides an interpretation of the data and presents the mechanism; Part V describes an application based on the test results; and Part VI is a summary, as shown in Figure 1.

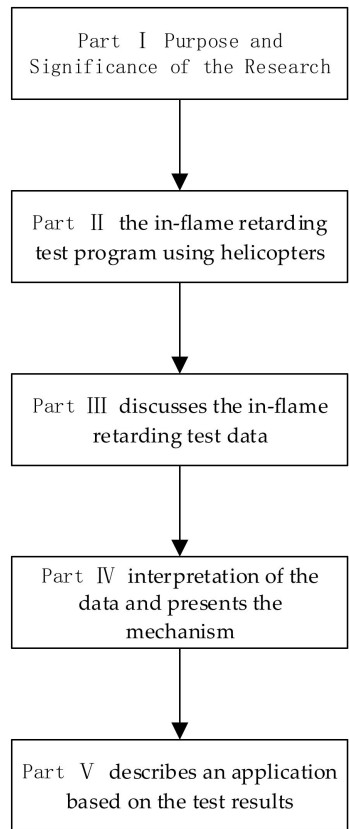

**Figure 1.** Flowchart for linking research parts.

## 2. Test Program

### 2.1. Test Combustibles

Wood cribs are often used for various tests requiring repeatable combustion temperatures, such as fire-extinguisher performance (ANSI/UL 711) [26,27]. Each crib was made of 1A wooden strips that were 40 mm × 40 mm × 500 mm in size, as stipulated in GB4351.1-2005 Portable Fire Extinguishers. The pine species *Pinus massoniana* Lamb was selected as the raw material for the wood strips. The size and layout of wood crib were all consistent with the literature [28] and are not described in detail in this paper and the 12-layer wood crib was shown in Figure 2.

The wood crib combustibles were placed on an iron mesh support that was 1200 mm × 1200 mm × 250 mm in size, and the ground under the wood crib combustibles was laid out with square fuel-oil basins that were 700 mm × 700 mm × 100 mm in size. Four liters of blue high-42 octane Avgas 100# aviation gasoline was used as the ignition fuel.

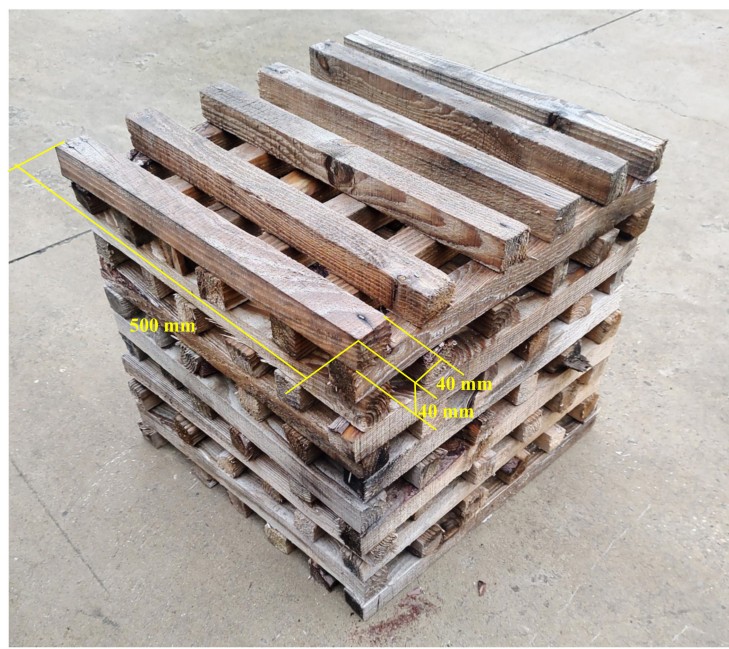

**Figure 2.** Photograph showing the construction of the 12-layer wood cribs.

## 2.2. Fire Scene Layout and Measurement Scheme

After the helicopter fire extinguishing device had sprayed the flame-retardant water agent at the same height, speed, and spraying conditions in the same trip, Crib#1, Crib#2, and Crib#3 were ignited after 1 h of natural drying to observe their burning characteristics. The main aim was to measure the flame retardant effect of the flame-retardant water agent sprayed by the helicopter fire extinguishing device. Among them, the structure size parameters, flight parameters, and distribution of the ground flame-retardant water agent released by the helicopter were all consistent with the literature [28] and are not described in detail in this paper. The radiation intensity [29], mass loss [30], and temperature [31] are regarded as the burning intensity features of wood crib fires. Therefore, this study mainly focused on the burning rate, radiation intensity, and temperature of the wood crib fire after the flame-retardant water agent was sprayed on the wood cribs. These features were used to characterize the intensity of combustion and to reflect the differences in the flame retardant effect of each agent on the wood cribs.

The tests used K-type thermocouples, the diameter of each thermocouple was 3 mm, the probe was bare, and the response time was one second. The uncertainty in the temperature of the Type-K thermocouple wire is given by the manufacturer as $\pm 2.2\ ^{\circ}$C with a 95% confidence interval. The expanded uncertainty for the thermocouple when the temperature change is from 0 $^{\circ}$C to 1250 $^{\circ}$C is 1.5% and the change from $-200\ ^{\circ}$C to 0 $^{\circ}$C is 4.0% with a coverage factor of 2, which corresponds to a confidence interval of 95%.

The positions of the thermocouple arrays are marked as Array#1, Array#2, and Array#3 in Figure 3. The positions and heights of thermocouples for measuring the temperature of 4-layer wood cribs, 12-layer wood cribs and 6-layer wood cribs are shown in Figures 4–6, respectively. Inside the wood crib, the thermocouples were relatively densely arranged with one thermocouple set on every two layers with a height difference of 0.08 m (the height of two layers of wooden strips). Two thermocouples were set on the upper part of the wood crib at a height of 1.0 m and 1.7 m.

An electronic scale was placed under the fire field of Array#1, Array#2, and Array#3. The electronic scale under the wood crib was used to measure the mass changes to the wood crib fire site during the burning process to obtain the mass loss rate of the wood cribs. The size of the electronic scale was 1200 mm × 1200 mm with a measuring range of 300 kg, division values of 100 g, and an accuracy of $\pm 0.02$%. The data were directly recorded by U disk, and the recording frequency was 1 time/second.

The tests used a Model C-3500 flame intensity calorimeter (America ITI company, USA) to obtain the heat fluxes. It was designed to be inserted directly into a flame front for the instantaneous determination of impinging heat energy. The length of the probe was 3 m and the probe diameter was 25.4 mm. The response time was less than 0.1 s, the max flux density was 31.9 kW/m$^2$, maximum operating temperature was 1920 °C nominal sensitivity was 314 w/m$^2$.uV, and the accuracy was 5%. The Portable Handheld Data Logger was a DaqPRO 5300 with a recording frequency of 1 time/second. There was only one ITI Model C-3500 flame intensity calorimeter in the laboratory, so it was installed at a height of 10 cm above the top surface of wood Crib#2, which was where the fire field intensity was greatest. The arrangement positions and heights of the thermocouple arrays, radiation heat flow meters, and the electronic scales are shown in Table 1.

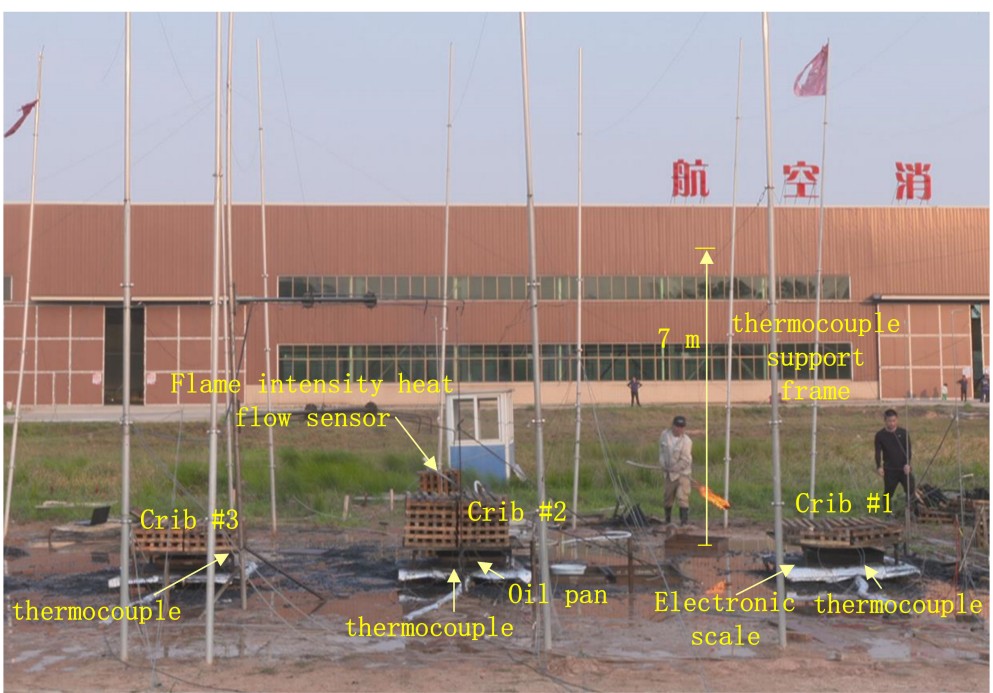

**Figure 3.** Actual layout of the flame-retardant test and measurements of temperature, radiant heat flux, and weight.

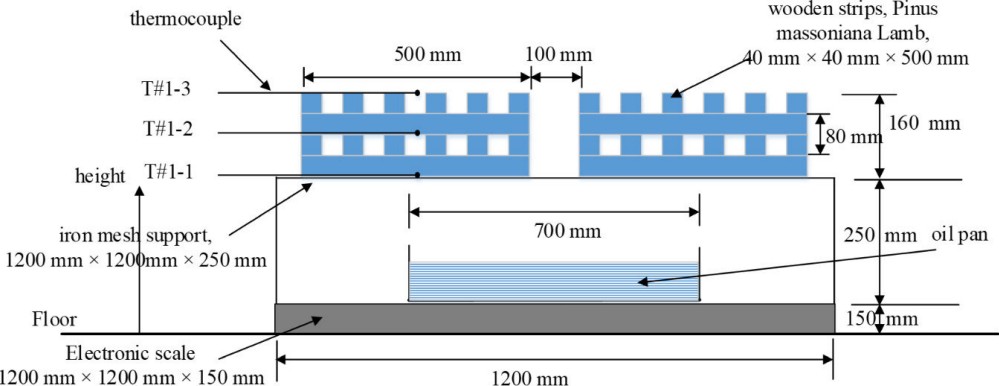

**Figure 4.** Diagram of the 4-layer wood crib apparatus.

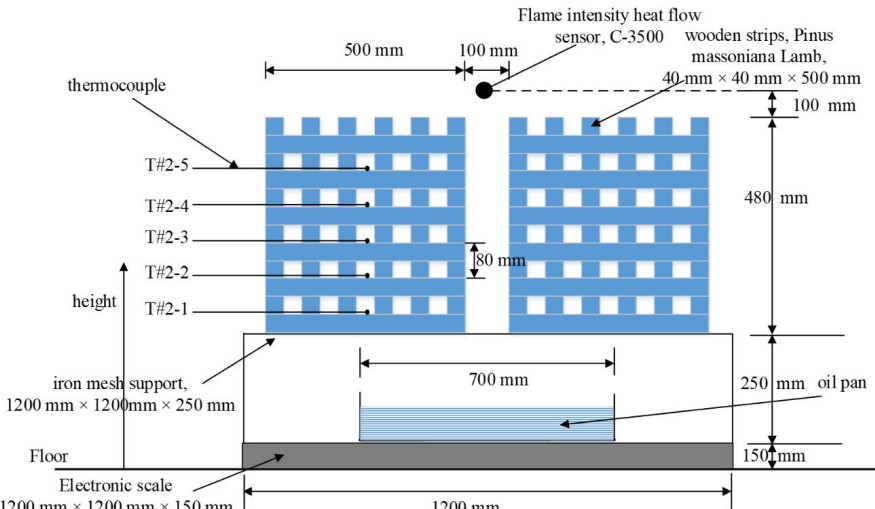

**Figure 5.** Diagram of the 12-layer wood crib apparatus.

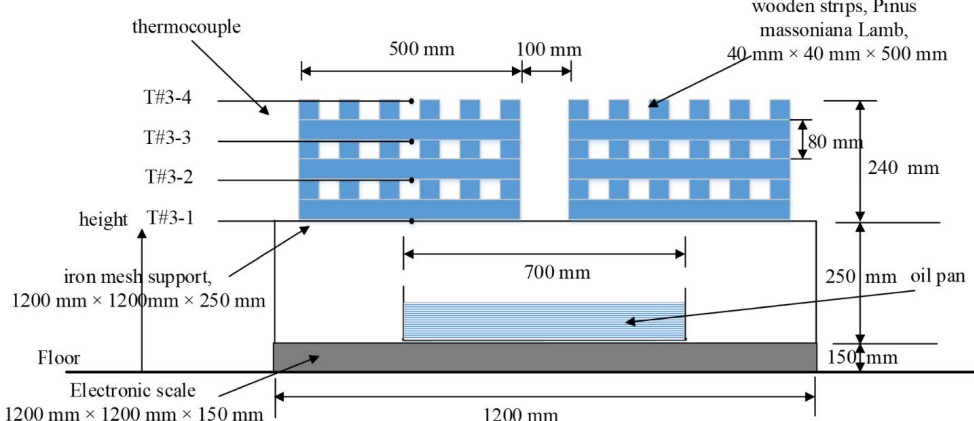

**Figure 6.** Diagram of the 6-layer wood crib apparatus.

**Table 1.** Layout of the thermocouples/scale and the flame intensity calorimeter in the fire retardants experiment.

| Serial No. | Thermocouple No. | Measuring Height/m | Inside or Above the Wood Crib | Location No. |
|---|---|---|---|---|
| 1 | Scale#1 | 0.0 | Underneath the wood crib | |
| 2 | T#1-1 | 0.25 | Inside the wood crib | |
| 3 | T#1-2 | 0.33 | Inside the wood crib | Array#1 |
| 4 | T#1-3 | 0.41 | Inside the wood crib | |
| 5 | T#1-4 | 1.0 | Above the wood crib | |
| 6 | T#1-5 | 1.7 | Above the wood crib | |
| 7 | Scale#2 | 0.0 | Underneath the wood crib | |
| 8 | T#2-1 | 0.29 | Inside the wood crib | |
| 9 | T#2-2 | 0.37 | Inside the wood crib | |
| 10 | T#2-3 | 0.45 | Inside the wood crib | |
| 11 | T#2-4 | 0.53 | Inside the wood crib | |
| 12 | T#2-5 | 0.61 | Inside the wood crib | Array#2 |
| 13 | T#2-6 | 1.0 | Above the wood crib | |
| 14 | T#2-7 | 1.7 | Above the wood crib | |
| 15 | T#2-8 | 2.7 | Above the wood crib | |
| 16 | RadiationHeatFlowMeters | 0.98 | Above the wood crib | |
| 17 | Scale#3 | 0.0 | Underneath the wood crib | |
| 18 | T#3-1 | 0.25 | Inside the wood crib | |
| 19 | T#3-2 | 0.33 | Inside the wood crib | |
| 20 | T#3-3 | 0.41 | Inside the wood crib | Array#3 |
| 21 | T#3-4 | 0.48 | Inside the wood crib | |
| 22 | T#3-5 | 1.0 | Above the wood crib | |
| 23 | T#3-6 | 1.7 | Above the wood crib | |

Three cameras were used during the test. Two were vertical to the fire field. Their short focal length lens was used to record whether the area covered by the flame-retardant water agent completely covered the three wood cribs in the flight direction. The long focal length camera and was used to record the combustion process after the flame-retardant water agent was sprayed on them. The third camera was parallel to the fire field to record whether the agent sprayed on the area completely covered the wood cribs in the perpendicular direction to the flight.

### 2.3. Fire Extinguishing Schemes

Pure water (Agent#1), 10% Class AB flame retardant (Agent#2), 0.3% gel flame retardant (Agent#3), 10% Class A flame retardant (Agent#4), and 10% Class A flame retardant + 0.6% guar gum (Agent#5) were used as the flame-retardant media to compare their flame-retardant properties when applied with a helicopter. The main components and descriptions of the various flame-retardant media are shown in Table 2.

**Table 2.** Main components of the flame-retardant water agents.

| No. | Flame Retardant Type | Weight Ratio of Flame Retardant to Water | Flame Retardant No. | Flame Retardant Components |
|---|---|---|---|---|
| 1 | Control group | | | The control group is not sprayed with any water agent |
| 2 | Pure water | / | Agent#1 | Pure water |
| 3 | Class AB flame retardant | 10% | Agent#2 | Main components: 0.1~2% fs-1157 fluorocarbon surfactant; 1~3% surface active betaine (α-Dodecyldimethyl betaine); 0.5~2% corrosion inhibitor (1H-Benzotriazole); The rest is water |
| 4 | Gel flame retardant | 0.3% | Agent#3 | Main components: Benzoin-SA Complex 50%; Polyacrylamide 50%; |
| 5 | Class A flame retardant | 10% | Agent#4 | Main components: 23 wt.% potassium chloride; 52~65 wt.% ammonium carbonate; 12~25 wt.% disodium hydrogen phosphate; The rest are flame retardants adjusted to a specific gravity of 1.1 by water |
| 6 | Class A flame retardant + 0.6% guar gum | 10% | Agent#5 | Main components: 23 wt.% potassium chloride; 52~65 wt.% ammonium carbonate; 12~25 wt.% disodium hydrogen phosphate; 0.6 wt.% Guar gum; The rest are flame retardants adjusted to a specific gravity of 1.1 by water |

Adding a thickener to the fire extinguishing agent can improve the wind resistance of the fire extinguishing agent and make it easier to adhere to the surface of the wooden pile, thereby improving the utilization rate of the fire extinguishing agent. Guar gum is a simple and readily available thickener. Therefore, this study applied guar gum to the flame-retardant water agents. The viscosity of the fire extinguishing agent at 0.6 wt.% was 190 mPa.s, which satisfies the above two conditions for improving the utilization rate.

The viscosities of Agent#3, Agent#4, and Agent#5 were tested and curves of the viscosity values for the fire extinguishing agents as a function of temperature were obtained, as shown in Figure 7. The viscosity of Agent#5 which mixed with guar gum decreased with the increase in temperature. The viscosities of Agent#3 and Agent#4 fire extinguishing agents without guar gum hardly changed with the increase in temperature. When the temperature of the Agent#5 fire extinguishing agent was lower than 20 °C, the viscosity of the fire extinguishing agent reached 190 mPa.s, but the viscosity dropped sharply when the temperature of the fire extinguishing agent was between 40 °C and 60 °C. When the temperature of the fire extinguishing agent reached 60 °C, its viscosity dropped to 55 mPa.s. It can be seen that the viscosity of Agent#5, when mixed with guar gum, was very sensitive to temperature.

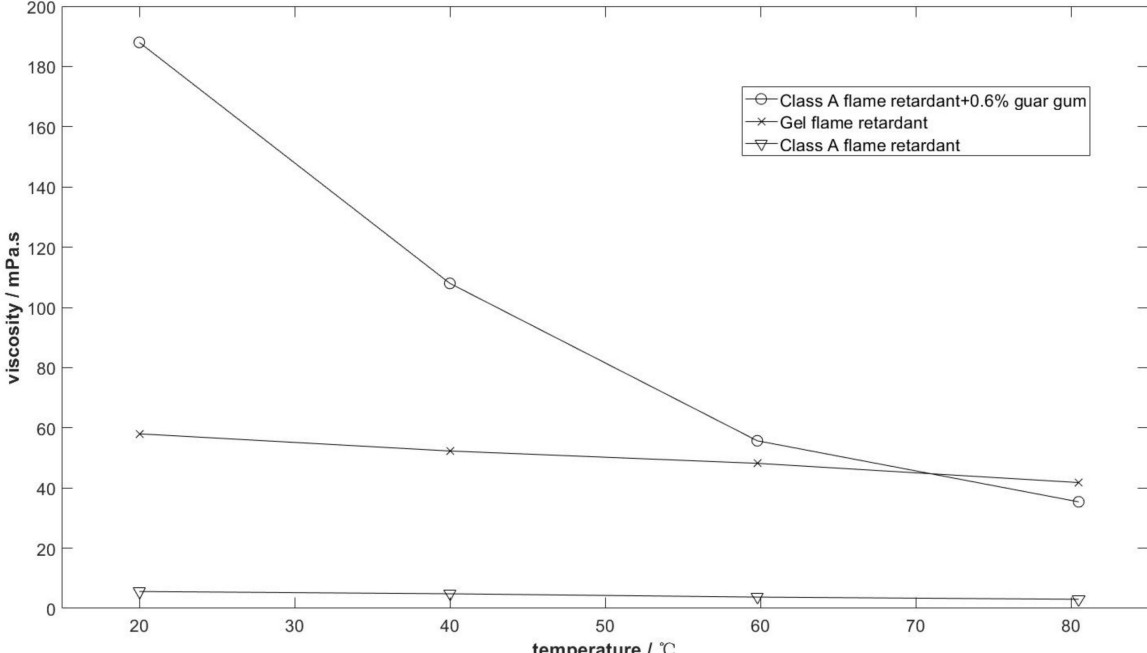

**Figure 7.** Viscosity of the fire extinguishing agents.

The test for flame retardancy by helicopter was divided into five steps:

(1) The helicopter carried the bucket fire extinguishing device to spray the flame-retardant water agent on the wood cribs three times at a speed of 20 km/h and at a height of 32 m, as shown in Figure 8;

(2) The wood cribs were allowed to naturally air-dry for 1 h;

(3) A certain amount of water and oil was added to the pilot oil pan to ignite the pilot oil under the wood cribs;

(4) The temperature, radiant heat flux density, and mass loss in the fire field were measured;

(5) A ground water pipe was used to extinguish the remaining fire in the wood crib when the Crib#2 wood crib fire field collapsed.

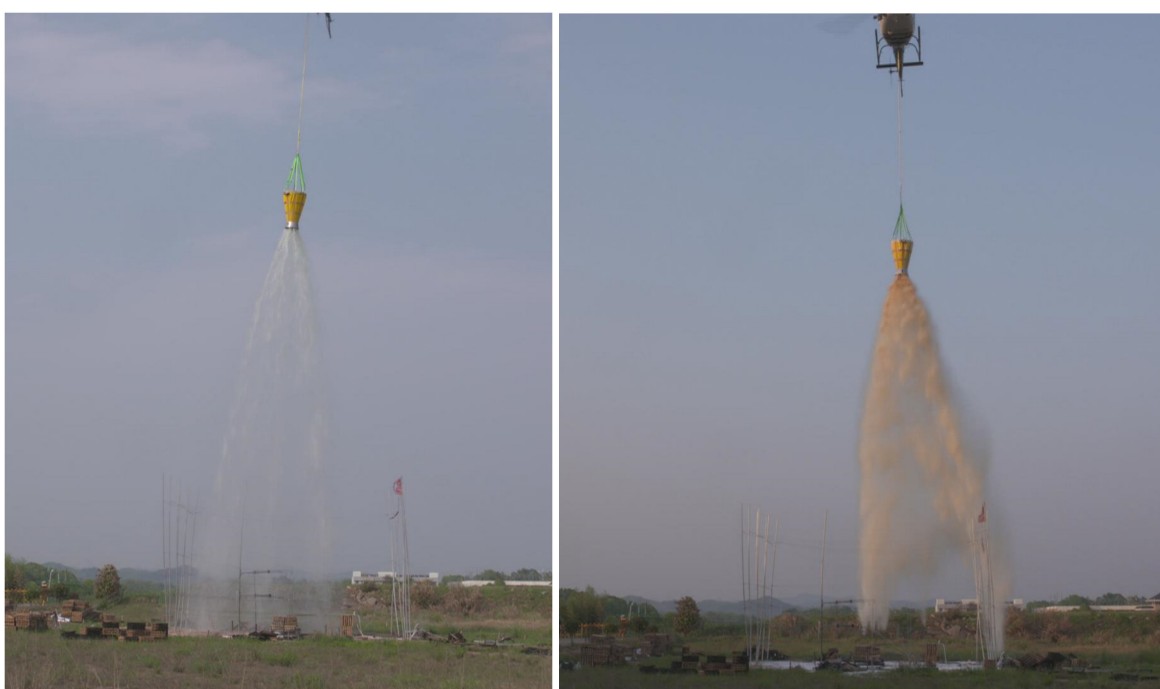

**Figure 8.** Helicopter high-altitude spraying of flame retardants. (**Left**): gel fire-extinguishing agent; (**Right**): class AB fire-extinguishing agent.

## 3. Flame Retardant Test

### 3.1. Comparative Testing of Flame Retardancy When Different Wood Crib Sizes Were Used

Mass Loss, Radiation Intensity and Temperature of the Wood Crib in the Control Group

Direct burning was performed on the surface of the wood crib where no spraying flame-retardant water agent was applied, and the mass loss, temperature, and radiant heat flux were measured after the wood crib was ignited. This treatment was the control group and was used to compare the differences in the parameters after helicopter spraying the flame-retardant water agents listed in Table 2.

Figure 9 shows the fire field burning conditions of the wood crib fire model when no flame-retardant water agent was applied. Screenshots at three time points were selected to show the burning conditions of the wood crib fire. These time points were 300 s (after burning of the wood stack pilot oil pan), 600 s (during the stable combustion stage), and 870 s (the minimum collapse time of the wood stack under the six flame retardant conditions).

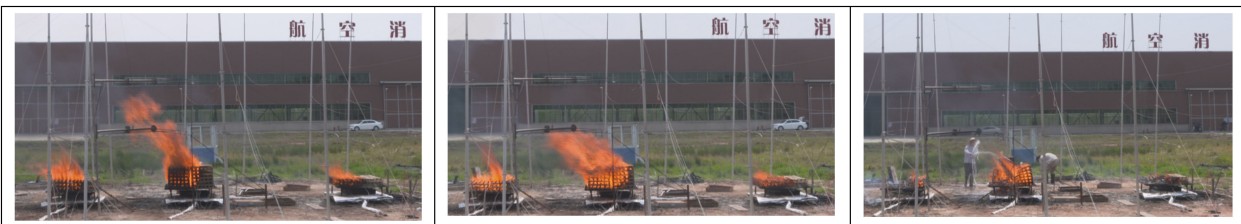

**Figure 9.** Combustion conditions of the fire scenes in the model without spraying flame retardants onto the wood crib. First subgraph: 180 s; second subgraph: 600 s; third subgraph: 870 s.

The mass loss, radiant heat flux density, and temperature change curves for the crib without any flame-retardant water agent are shown in Figure 10. The change curves for temperature, radiant heat flux density, and the mass of the wood cribs are represented by three subgraphs. The three subgraphs from top to bottom represent the changes process for Crib#1, Crib#2, and Crib#3, respectively. The maximum mass loss refers to the average value of the mass loss over 5 consecutive seconds, and the average value of the radiant

heat flux value refers to the average value of the radiant heat flux density of Crib#2 from the burnt of the oil to the collapse of the wood crib. This was to avoid the influence of any mutation values caused by interference factors during the data acquisition process. It can be seen from the mass loss, radiant heat flux density, and temperature curves that after the pilot oil pan was ignited, the 100# aviation gasoline in the oil pan and the wood crib started to burn violently, releasing large amounts of heat. The temperature and radiant heat flux density curves increased rapidly and the quality gradually decreased, reaching a peak at about 90 s. After about 180 s, as the pilot fuel in the oil pan was consumed, the temperature measured by the thermocouples at all heights dropped and the wood crib entered the free combustion stage.

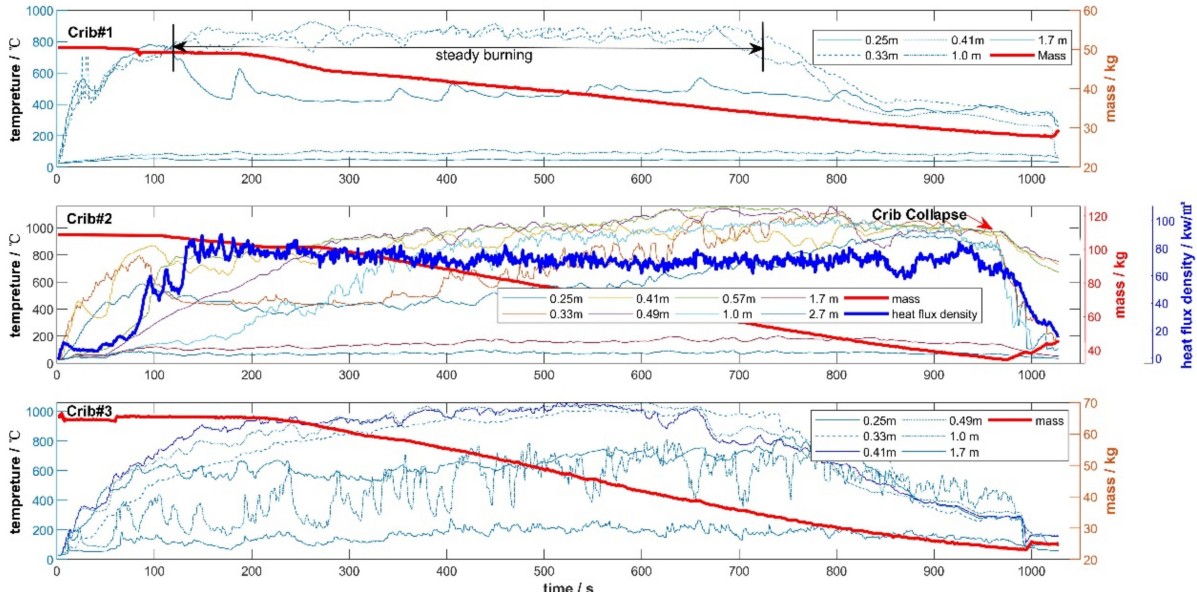

**Figure 10.** Temperature, radiant heat flux density, and mass change curves for the wood cribs not sprayed with fire extinguishing agent.

The temperature curve for the Crib#1 wood crib fire was in the stable combustion stage from 180 to 720 s, with a maximum temperature of 928 °C, and was when the wood crib mass loss was most rapid. The average wood crib mass loss during the stable combustion stage was 0.026 kg/s. After 720 s of the test, the combustion entered the decay stage, the temperature measured by the thermocouples gradually decreased, and the rate of mass decline eased.

The temperature curve for the Crib#2 wood crib fire was in the stable combustion stage from 200 to 800 s, with a maximum temperature of 1156 °C. The radiant heat flux value reached 110 kW/m$^2$, which was when the wood crib mass loss was most rapid. The average wood crib mass loss in the stable combustion stage was 0.115 kg/s. After 1360 s, the combustion entered the decay stage, the temperature measured by the thermocouple gradually decreased, and the rate of mass decline eased. After 2106 s, a small number of the wood strips fell off due to the loss of supporting force (carbonization) and the crib lost its original shape.

The temperature curve for the Crib#3 fire was in the stable combustion stage from 180 to 740 s and reached a maximum temperature of 1056 °C, which was when the wood crib mass loss was most rapid. The average wood crib mass loss in the stable combustion stage was 0.053 kg/s. After 740 s, the combustion entered the decay stage, the temperature measured by the thermocouple gradually decreased, and the rate of mass decline eased.

The data measured by the thermocouples and radiant heat flux density meter after the collapse of the wooden crib was no longer the measurement value of the assumed position in the design plan so ground water pipes were used to extinguish the residual fire associated with the wooden cribs after their collapse. At this time, the temperatures of the

Crib#1, Crib#2, and Crib#3 fires decreased rapidly, their quality increased rapidly under the effect of water spraying from the ground, and their data was of no practical significance. However, due to the limited number of test personnel, when ground water pipes were used to extinguish the residual wood crib fire, water was not sprayed on Crib#1, Crib#2, and Crib#3 fires at the same time. This resulted in a rapid drop in temperature and a rapid mass increase at the three fire sites that did not occur at the same time. This is reflected in the subsequent analysis charts.

### 3.2. Tests on Flame-Retardant Water Agents When Sprayed from a Helicopter

3.2.1. Coverage by the Flame-Retardant Water Agent When Sprayed from a Helicopter

In order to intuitively describe the coverage of the flame-retardant water agent on wood crib fires when a helicopter sprayed the flame-retardant water agent on three separate occasions, the flame-retardant agent has been shown overlapping the water agent applied to Crib#1, Crib#2, and Crib#3, as shown in Figure 11. To better record the track of the wood crib and flame-retardant water agent, the coverage area of flame-retardant water agent was assumed to be approximately oval, and the size of wood crib has been enlarged. The sub-graphs showing the coverage by each agent are in the following order from top to bottom: Crib#1, Crib#2, and Crib#3 in each subgraph.

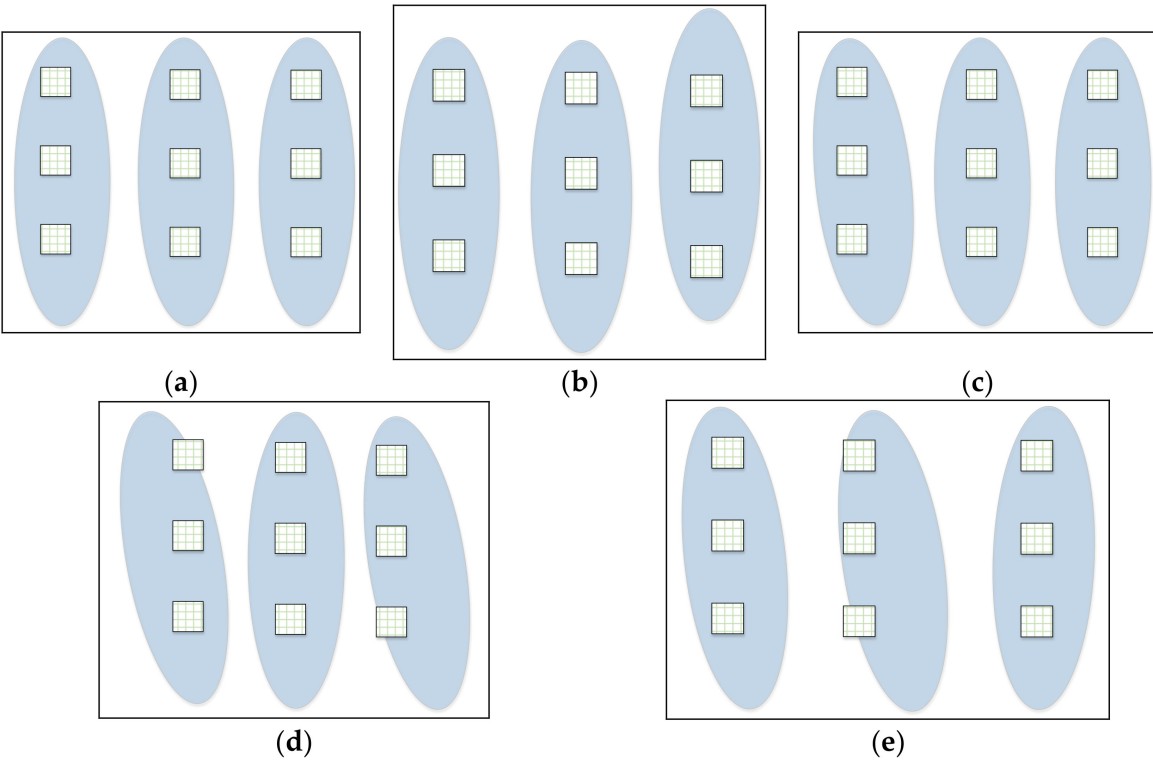

**Figure 11.** Coverage areas for each retardant agent dropped from the helicopter: (**a**) Agent#1, (**b**) Agent#2, (**c**) Agent#3, (**d**) Agent#4, and (**e**) Agent#5. Crib#1, Crib#2, Crib#3 from top to bottom.

It can be seen from Table 3 that the weight increases due to the Class AB flame retardant and the gel flame retardant are larger than those for the other flame-retardant water agents and more of the flame-retardant water agents are absorbed. This is mainly related to the active ingredients in Class AB flame retardants. The flame-retardant water agent can quickly penetrate the surface of the wood crib and enter the interior. The gel flame retardant has poor fluidity, good adhesion performance, and can adhere to the surface of the wood crib. Therefore, the loss of the flame-retardant water agent with gel is less than that of other flame-retardant water agents.

**Table 3.** Weight changes after flame-retardant water agents were sprayed by a helicopter.

| No. | Flame Retardant | Crib#1 Initial Weight/kg | Crib#1 Weight after Spraying Fire-Extinguishing Agent/kg | Crib#2 Initial Weight/kg | Crib#2 Weight after Spraying Fire-Extinguishing Agent/kg | Crib#3 Initial Weight/kg | Crib#3 Weight after Spraying Fire-Extinguishing Agent/kg |
|---|---|---|---|---|---|---|---|
| 1 | Control Group | 42.3 | / | 115.5 | / | 57.2 | / |
| 2 | Pure Water | 40.5 | 46.2 | 121.6 | 129.75 | 58.3 | 67.9 |
| 3 | Class AB Flame Retardant | 40.2 | 47.35 | 117.4 | 122.4 | 59.6 | 66.65 |
| 4 | Gel Flame Retardant | 39.9 | 55.25 | 126.3 | 148.1 | 58.5 | 68.4 |
| 5 | Class A Flame Retardant | 38.9 | 42.1 | 120.1 | 126 | 55.6 | 56.9 |
| 6 | 0.6% Thickened Class A Flame Retardant | 41.3 | 47 | 125.8 | 138.2 | 58.0 | 66.1 |

### 3.2.2. Pure Water (Agent#1)

The pure water (Agent#1) flame retardant test was carried out at a temperature of 29 °C. Before spraying the pure water (wind speed generated by the helicopter rotor was not considered), the wind speed was 2.1 m/s. The natural wind speed and humidity when the wood cribs were ignited were 2.4 m/s and 52%, respectively. Figures 12 and 13 show the fire field burning conditions of the helicopter bucket using Agent#1 to protect the wood crib fire model.

Starting from 150 s, the fire field maintained a relatively stable combustion level for about 180 s. At this stage, wood crib combustion was relatively stable, there was uniform heat release, and the wood crib mass loss increased.

The temperature curve for the Crib#1 fire was in the stable combustion stage from 250 to 1100 s with a maximum temperature of 623 °C, which was when the wood crib mass loss was most rapid. The average wood crib mass loss in the stable combustion stage was 0.0160 kg/s.

The temperature curve for the Crib#2 fire was in the stable combustion stage from 250 to 1100 s, with a maximum temperature of 1110 °C. The radiant heat flux value reached 114 kW/m$^2$, the wood crib mass loss was most rapid at this point, and the average wood crib mass loss during the stable combustion stage was 0.1220 kg/s. After 1100 s, the combustion entered the decay stage, the temperature measured by the thermocouple gradually decreased, and the rate of mass decline eased. At 1120 s after igniting the wood crib, a small number of the wood strips fell off due to the loss of supporting force (carbonization) and the wood crib lost its original shape.

The temperature curve for the Crib#3 fire was in the stable combustion stage from 250 to 1050 s. It reached a maximum temperature of 917 °C when the wood crib mass loss reached its peak. The average wood crib mass loss in the stable combustion stage was 0.0432 kg/s. At 1050 s after igniting the wood crib, the combustion entered the decay stage, the temperature measured by the thermocouple gradually decreased, and the rate of mass decline eased.

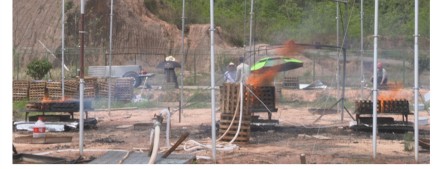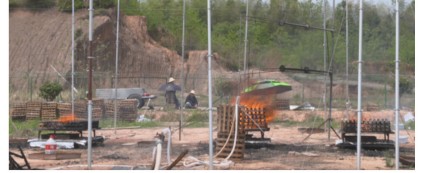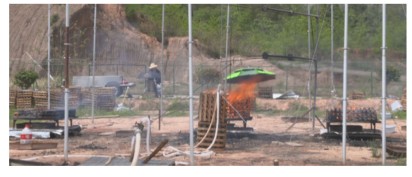

**Figure 12.** Combustion conditions of the fire scenes in the model using Agent#1 taken from the helicopter after spraying flame retardants onto the wood crib. First subgraph: 180 s; second subgraph: 600 s; third subgraph: 870 s.

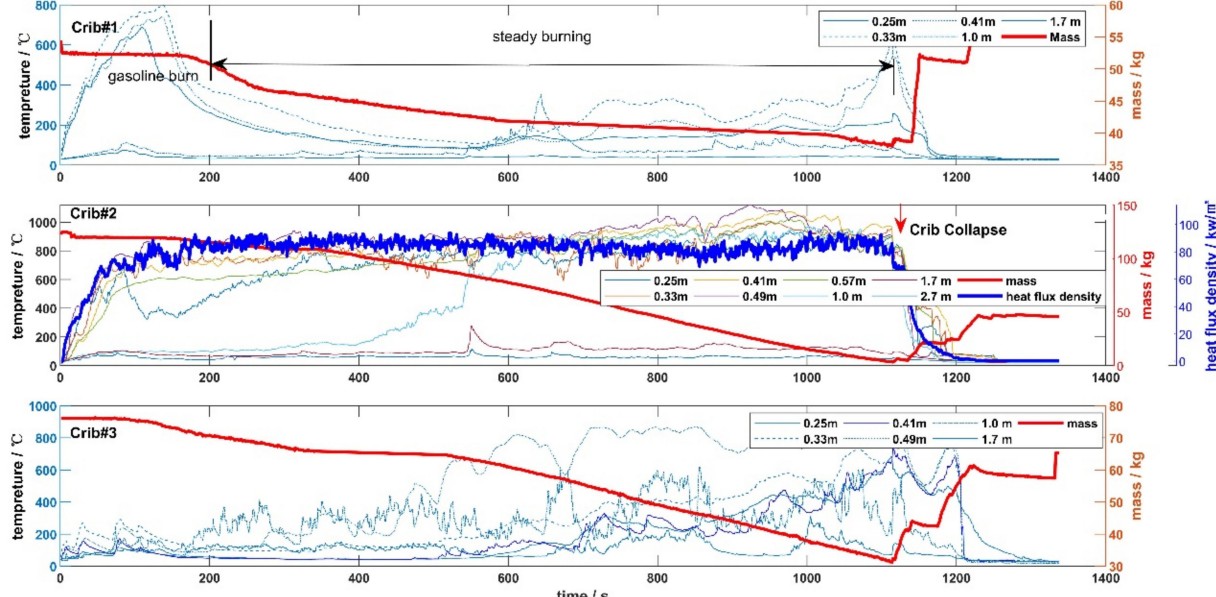

**Figure 13.** Temperature, radiant heat flux density, and mass change curves for the wood cribs after the helicopter sprayed Agent#1 fire extinguishing agent.

### 3.2.3. Class AB Flame Retardant (Agent#2)

The flame-retardant test of the Class AB flame retardant (Agent#2) was carried out at 32 °C. Before spraying the flame-retardant water agent (wind speed generated by the helicopter rotor was not considered), the wind speed was 2.4 m/s. The natural wind speed and humidity natural wind speed when the wood cribs were ignited were 1.8 m/s and 56%, respectively. Figures 14 and 15 show the fire field burning conditions after the helicopter had sprayed Agent#2 flame-retardant water agent to protect the wood crib model. The strong wind generated by the helicopter rotor when applying the flame retardant and during flight meant that the flame-retardant water agent became foam and dissipated. Therefore, the amount of the agent applied was less than that for the other flame retardants. In addition, Agent#2 covered a wider area than Agent#1, Agent#3, and Agent#4 when carrying out firefighting, which meant that the depth of the flame-retardant water agent per unit area was smaller.

Class AB flame retardant is used for oil fire extinguishing and as a flame retardant. This means that when Class AB flame retardant is sprayed by helicopter onto the combustion plate, the wood crib ignition process can be hindered. Therefore, 100# aviation gasoline was added after all the Class AB flame retardant remaining in the combustion plate had been poured out.

The temperature curve for the Crib#1 fire was in the stable combustion stage from 300 to 1186 s. It reached a peak temperature of 887 °C when the wood crib mass loss was at its most rapid. The average wood crib mass loss during the stable combustion stage was 0.023 kg/s. At 1186 s after igniting the wood crib, the combustion entered the decay stage,

the temperature measured by the thermocouple gradually decreased, and the rate of mass decline eased.

The temperature curve for the Crib#2 fire was in the stable combustion stage from 300 to 1360 s. It reached a peak temperature of 1067 °C. At this point, the ignition process of the radiant heat flux value reached 100 kW/m$^2$ and the wood crib mass loss was most rapid. The average wood crib mass loss during the stable combustion stage was 0.1047 kg/s. After 1360 s, the combustion entered the decay stage, the temperature measured by the thermocouple gradually decreased, and the rate of mass decline eased. At 2106 s after igniting the wood crib, a small number of the wood strips fell off due to the loss of supporting force (carbonization), which meant that the wood crib lost its original shape.

The temperature curve for the Crib#3 fire was in the stable combustion stage from 300 to 1260 s. It reached a peak temperature of 990 °C, which was when the wood crib mass loss was most rapid. The average wood crib mass loss during the stable combustion stage was 0.0429 kg/s. At 1260 s after igniting the wood crib, the combustion entered the decay stage, the temperature measured by the thermocouple gradually decreased, and the rate of mass decline eased.

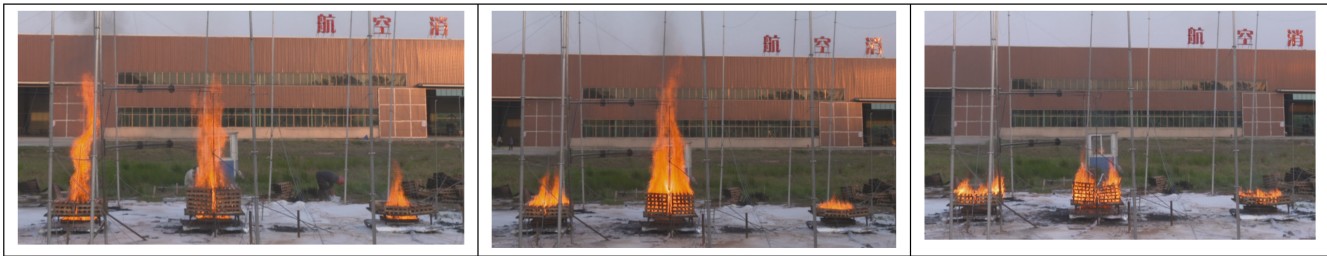

**Figure 14.** Combustion conditions of the fire scenes in the model using Agent#2 taken from the helicopter after spraying the flame retardant onto the wood crib. First subgraph: 180 s; second subgraph: 600 s; third subgraph: 870 s.

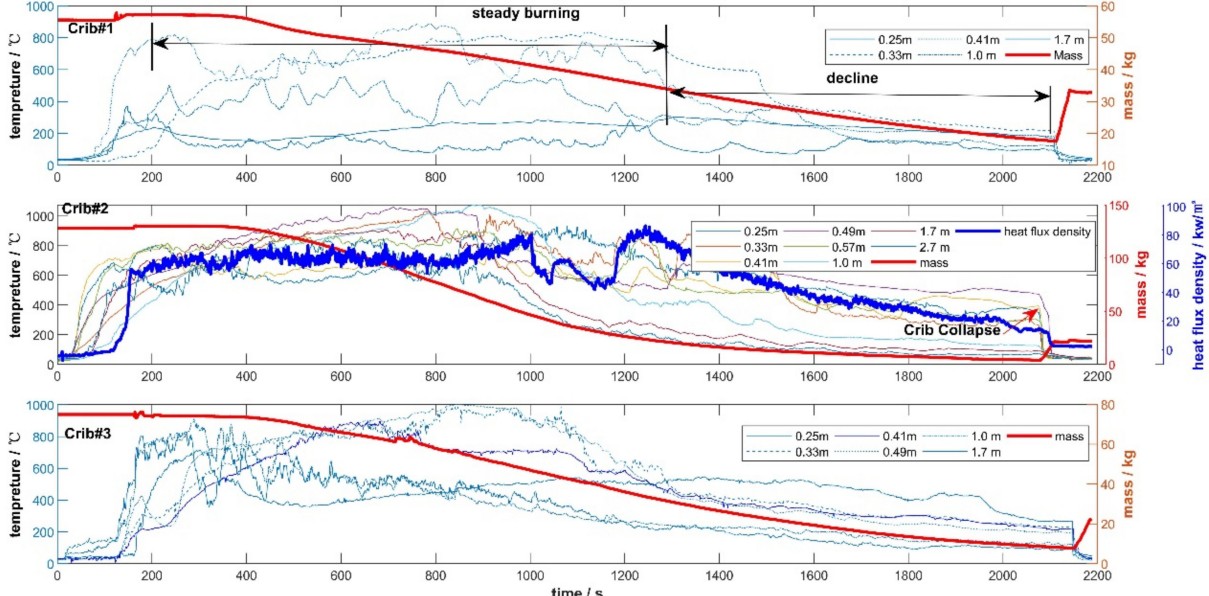

**Figure 15.** Temperature, radiant heat flux density, and mass change curves for the wood cribs after the helicopter had sprayed Agent#2 fire extinguishing agent.

3.2.4. Gel Flame Retardant (Agent#3)

The flame retardant test of gel flame retardant (Agent#3) was carried out at 29 °C. Before spraying the flame-retardant water agent (wind speed generated by the helicopter rotor was not considered), the wind speed was 1.9 m/s. The natural wind speed and

humidity natural wind speed when the wood crib was ignited were 2.2 m/s and 48%, respectively. Figures 16 and 17 show the fire field burning conditions after the helicopter sprayed the crib using Agent#3 to extinguish the non-uniform wood crib fire model.

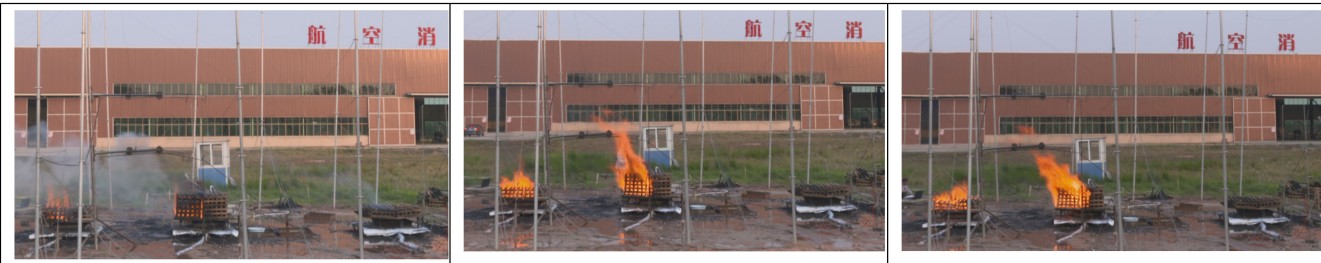

**Figure 16.** Combustion conditions of the fire scenes in the model using Agent#3 taken from the helicopter after spraying the flame retardant onto the wood crib. First subgraph: 180 s; second subgraph: 600 s; third subgraph: 870 s.

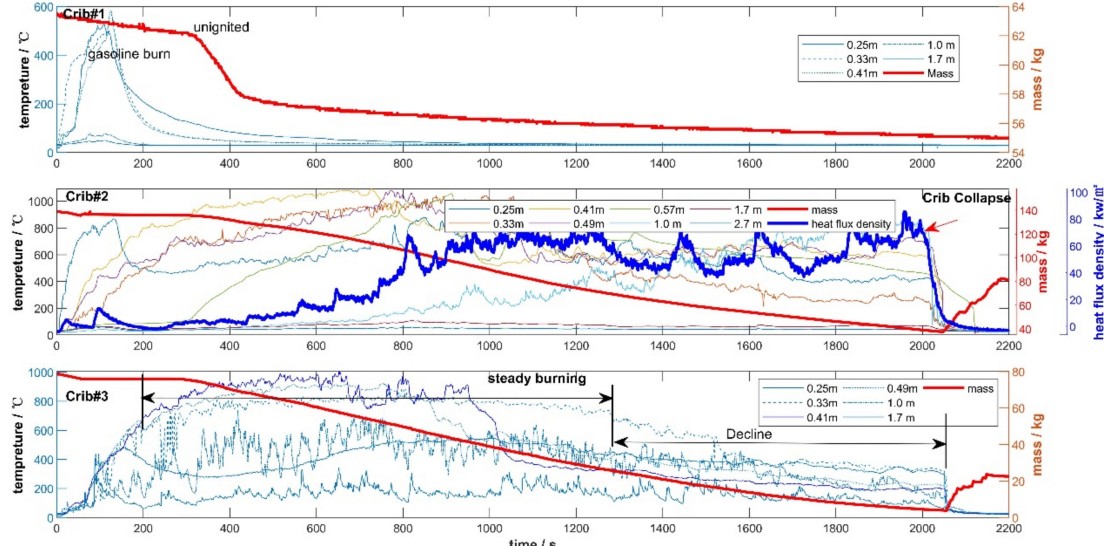

**Figure 17.** Temperature, radiant heat flux density, and mass change curves for the wood cribs after the helicopter had sprayed Agent#3 fire extinguishing agent.

The temperature curve for the Crib#1 fire was in the ignition stage from 0 to 127 s and the peak temperature was 583 °C. Since the wooden crib only had four layers, it could be completely covered by the gel sprayed by the helicopter. Thus, the wood crib did not ignite, the fire field was extinguished, and the pilot gasoline burned out after 127 s.

The temperature curve for the Crib#2 fire was in the stable combustion stage from 280 to 1340 s when a peak temperature of 1091 °C was reached. During this stage, the radiant heat flux value reached 102 kW/m² and the wood crib mass loss was most rapid. The average value of the mass loss of the wood crib in the stable combustion stage was 0.0799 kg/s. After 1500 s, the combustion entered the decay stage, the temperature measured by the thermocouple gradually decreased, and the rate of mass decline eased. At 2010 s after igniting the wood crib, a small number of the wood strips fell off due to the loss of supporting force (carbonization), which meant that the crib lost its original shape.

The temperature curve for the Crib#3 fire was in the stable combustion stage from 200 to 1250 s and reached a peak temperature of 997 °C, which was when the wood crib mass loss was most rapid. The average wood crib mass loss during the stable combustion stage was 0.0465 kg/s. At 1250 s after igniting the wood crib, the combustion entered the decay stage, the temperature measured by the thermocouple gradually decreased, and the rate of mass decline eased.

The gel flame retardant has good adhesion performance and can effectively prevent the wood cribs from burning when it adheres to the surface of the wood crib. In addition, Crib#1 had fewer layers and the gel flame retardant completely covered it, so that the wood crib did not ignite.

### 3.2.5. The 10% Class A Flame Retardant (Agent#4)

The flame-retardant test of 10% Class A flame retardant (Agent#4) was carried out at 31 °C. Before spraying the flame-retardant water agent (wind speed generated by the helicopter rotor was not considered), the wind speed was 1.6 m/s. The natural wind speed and humidity natural wind speed when the wood crib was ignited were 1.5 m/s and 44%, respectively. Figures 18 and 19 show the fire field burning conditions after the helicopter had sprayed the cribs using Agent#4 to extinguish the non-uniform wood crib fire model.

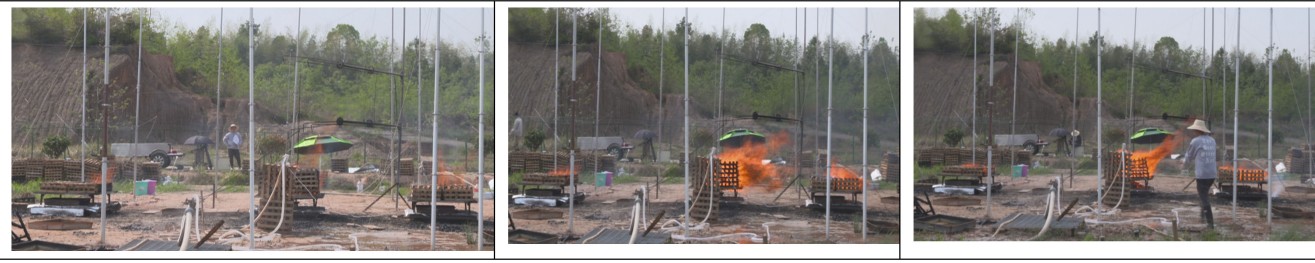

**Figure 18.** Combustion conditions of the fire scenes in the model using Agent#4 taken from the helicopter after spraying the flame retardants onto the wood crib. First subgraph: 180 s; second subgraph: 600 s; third subgraph: 870 s.

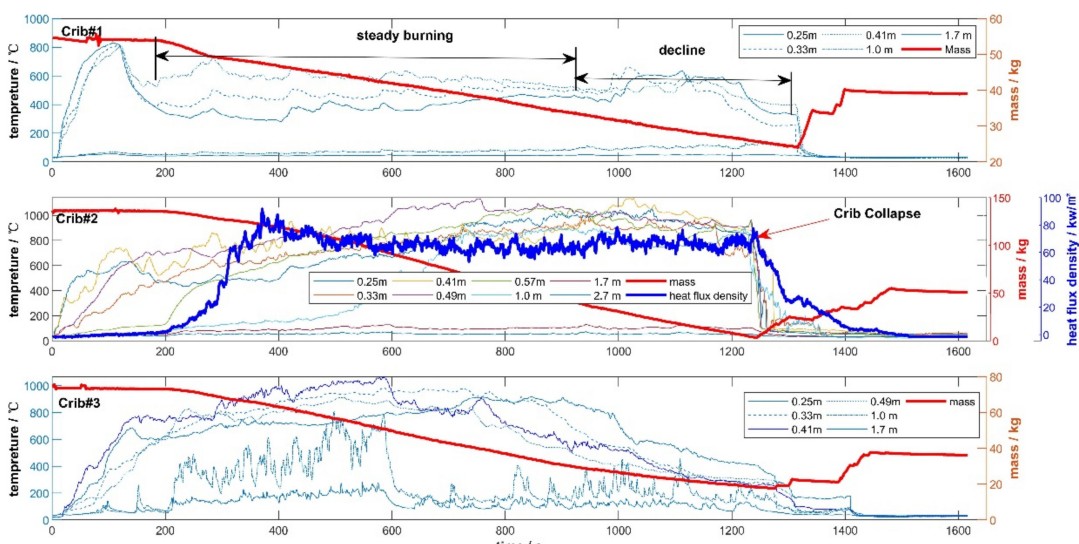

**Figure 19.** Temperature, radiant heat flux density, and mass change curves for the wood cribs after the helicopter had sprayed Agent#4 fire extinguishing agent.

The temperature curve of the Crib#1 fire was in the stable combustion stage from 200 to 900 s when it reached a peak temperature of 729 °C and the wood crib mass loss was most rapid. The average wood crib mass loss during the stable combustion stage was 0.0266 kg/s. At 900 s after igniting the wood crib, the combustion entered the decay stage, the temperature measured by the thermocouple gradually decreased, and the rate of mass decline eased.

The temperature curve for the Crib#2 fire was in the stable combustion stage from 300 to 1120s when it reached a peak temperature of 1128 °C. At this point, the radiant heat flux value reached 94 kW/m$^2$ and the wood crib mass loss was most rapid. The average value of the mass loss of the wood crib during the stable combustion stage was

0.0867 kg/s. After 1230 s, the combustion entered the decay stage, the temperature measured by the thermocouple gradually decreased, and the rate of mass decline eased. At 1230 s after igniting the wood crib, a small number of the wood strips fell off due to the loss of supporting force (carbonization), which meant that the crib lost its original shape.

The temperature curve for the Crib#3 fire was in the stable combustion stage from 200 to 920 s when it reached a peak temperature of 1058 °C and the wood crib mass loss was most rapid. The average wood crib mass loss during the stable combustion stage was 0.0522 kg/s. At 920 s after igniting the wood crib, the combustion entered the decay stage, the temperature measured by the thermocouple gradually decreased, and the rate of mass decline eased.

### 3.2.6. The 10% Class A flame Retardant + 0.6% Guar Gum (Agent#5)

The flame-retardant test of the 10% Class A flame retardant + 0.6% guar gum (Agent#5) was carried out at 30 °C. Before spraying the flame-retardant water agent (wind speed generated by the helicopter rotor was not considered), the wind speed was 1.8 m/s. The natural wind speed and humidity natural wind speed when the wood crib was ignited were 2.1 m/s and 51%, respectively. Figures 20 and 21 show the fire field burning conditions after the helicopter had sprayed the cribs using Agent#5 to extinguish the continuous and uniform wood crib fire model.

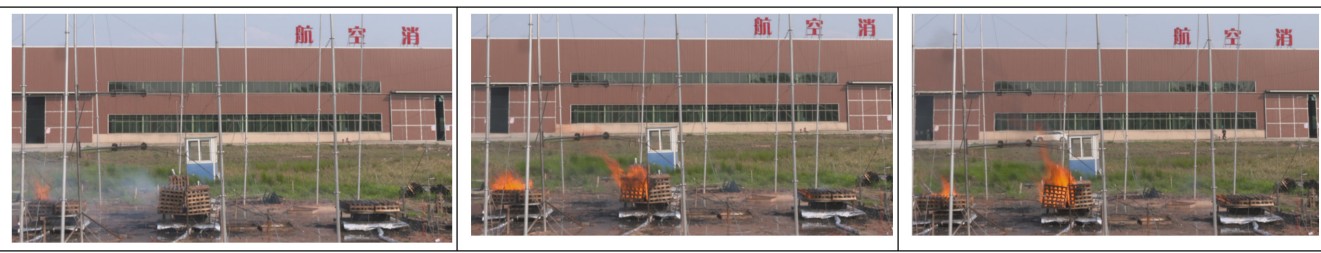

**Figure 20.** Combustion conditions of the fire scenes in the model using Agent#5 taken from the helicopter after spraying flame retardants onto the wood crib. First subgraph: 180 s; second subgraph: 600 s; third subgraph: 870 s.

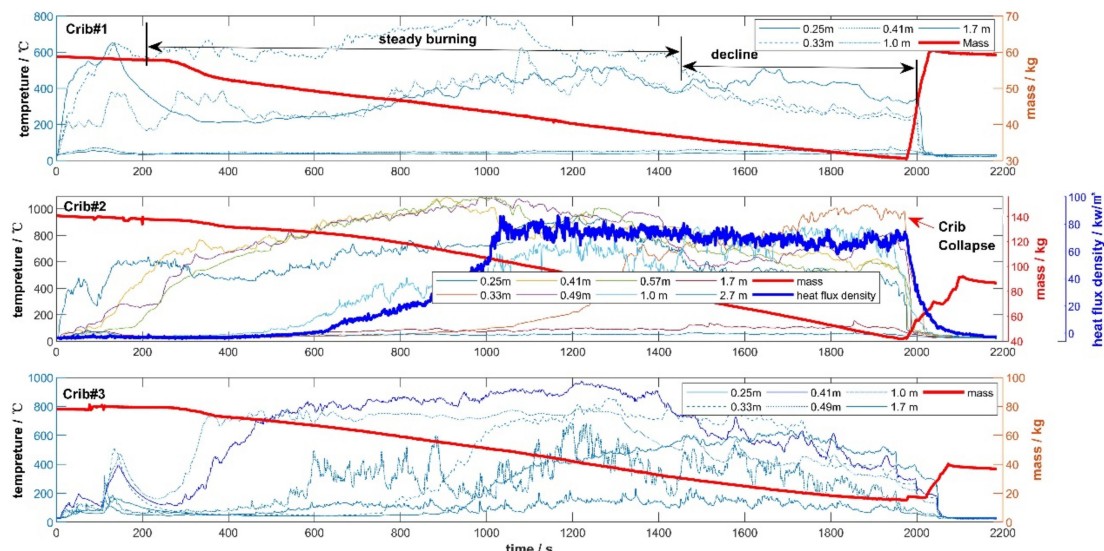

**Figure 21.** Temperature, radiant heat flux density, and mass change curves for the wood cribs after the helicopter had sprayed Agent#5 fire extinguishing agent.

The temperature curve for the Crib#1 fire was in the stable combustion stage from 250 to 1430 s when it reached a peak temperature of 798 °C and the wood crib mass loss was most rapid. The average wood crib mass loss during the stable combustion stage was

0.0222 kg/s. At 1430 s after igniting the wood crib, the combustion entered the decay stage, the temperature measured by the thermocouple gradually decreased, and the rate of mass decline eased.

The temperature curve for the Crib#2 fire was in the stable combustion stage from 250 to 1750 s when it reached a peak temperature 1080 °C. At this point, the radiant heat flux value reached 87 kW/m$^2$ and the wood crib mass loss was most rapid. The average value for wood crib mass loss in the stable combustion stage was 0.0630 kg/s. After 1750 s, the combustion entered the decay stage, the temperature measured by the thermocouple gradually decreased, and the rate of mass decline eased. At 1980 s after igniting the wood crib, a small number of the wood strips fell off due to the loss of supporting force (carbonization), which meant that the crib lost its original shape.

The temperature curve for the Crib#3 fire was in the stable combustion stage from 350 to 1480 s when it reached a peak temperature of 967 °C and the wood crib mass loss was most rapid. The average wood crib mass loss during the stable combustion stage was 0.0418 kg/s. At 1480 s after igniting the wood crib, the combustion entered the decay stage, the temperature measured by the thermocouple gradually decreased, and the rate of mass decline eased.

## 4. Discussion and Analysis

Table 4 and Figures 22–25 summarize the highest temperatures, highest radiant heat flux densities, the average radiant heat flux values, and the average mass losses during the stable combustion stage under the action of the different flame retardants (including no flame retardant sprayed).

**Table 4.** Summary of flame retardant performance indices of the wood crib fire sites.

| Crib No. | Project | Control Group | Agent#1 | Agent#2 | Agent#3 | Agent#4 | Agent#5 |
|---|---|---|---|---|---|---|---|
| Crib#1 | Maximum temperature (°C) | 928 | 623 | 887 | 583 | 729 | 798 |
| | Average mass loss in stable combustion stage (kg/s) | 0.026 | 0.0160 | 0.023 | / | 0.0266 | 0.0222 |
| Crib#2 | Maximum temperature (°C) | 1156 | 1110 | 1067 | 1091 | 1128 | 1080 |
| | The maximum value of radiant heat flux (kw/m$^2$) | 110 | 114 | 100 | 102 | 94 | 87 |
| | The average value of radiant heat flux (kw/m$^2$) | 87.95 | 101.56 | 74.61 | 43.14 | 63.9 | 45.77 |
| | Average mass loss in stable combustion stage (kg/s) | 0.115 | 0.1359 | 0.1047 | 0.0799 | 0.0867 | 0.0630 |
| Crib#3 | Maximum temperature (°C) | 1056 | 917 | 990 | 997 | 1058 | 967 |
| | Average mass loss in stable combustion stage (kg/s) | 0.053 | 0.0432 | 0.0429 | 0.0465 | 0.0522 | 0.0418 |

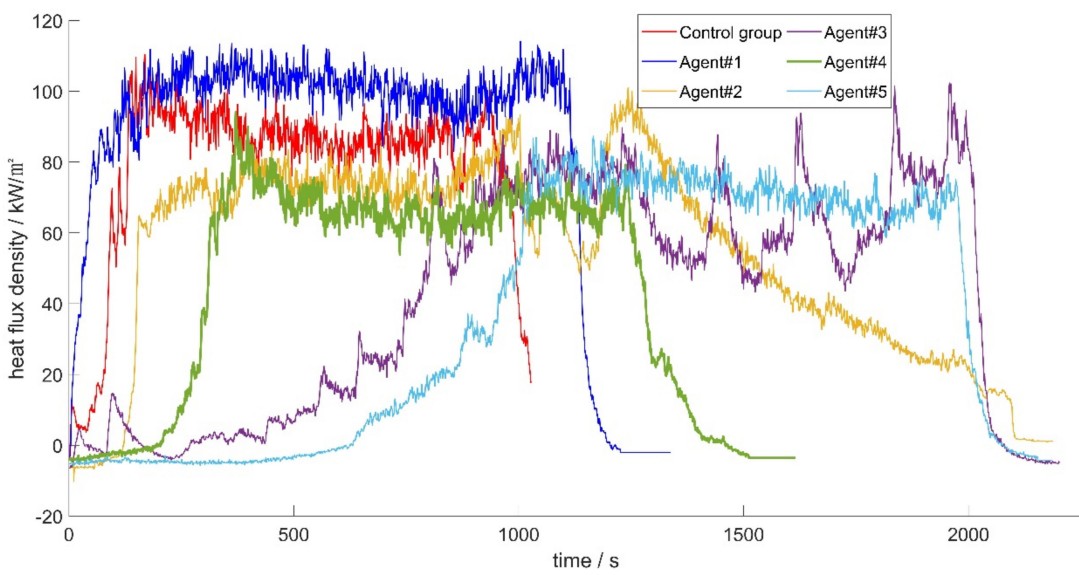

**Figure 22.** Differences in heat flux density after the helicopter had sprayed the flame retardants.

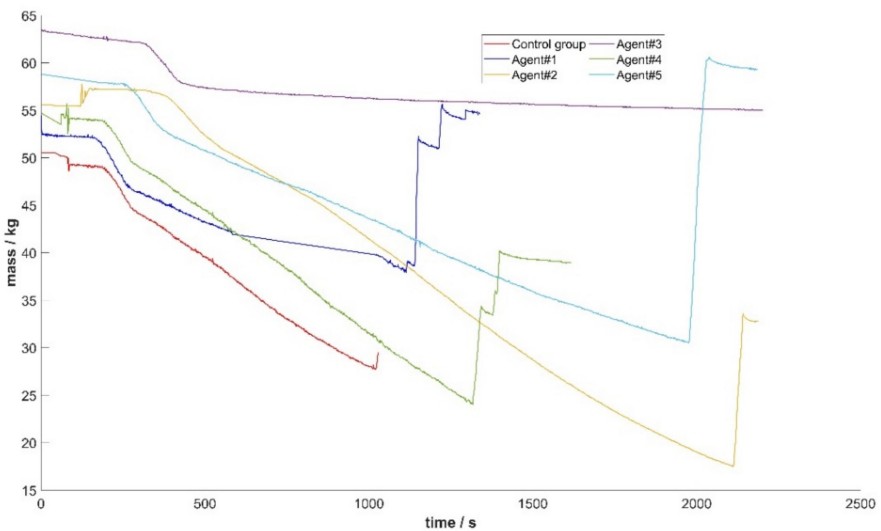

**Figure 23.** Differences in crib#1 mass losses after the helicopter had sprayed the flame retardants.

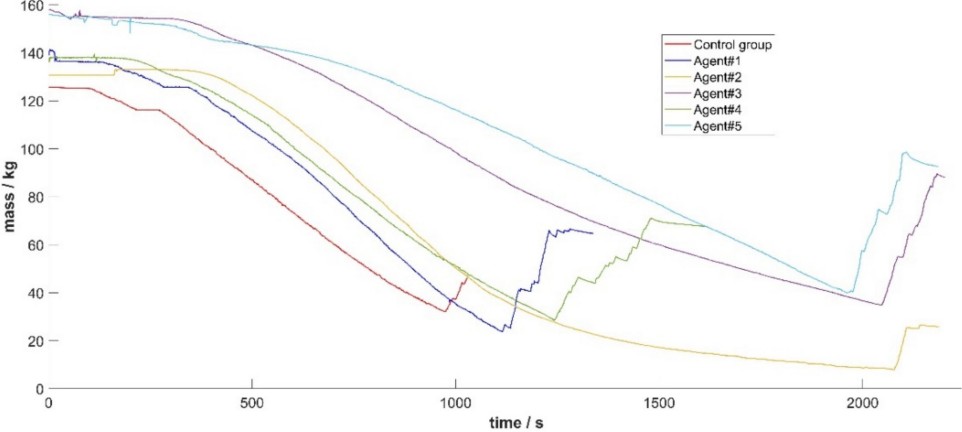

**Figure 24.** Differences in crib#2 mass losses after the helicopter had sprayed the flame retardants.

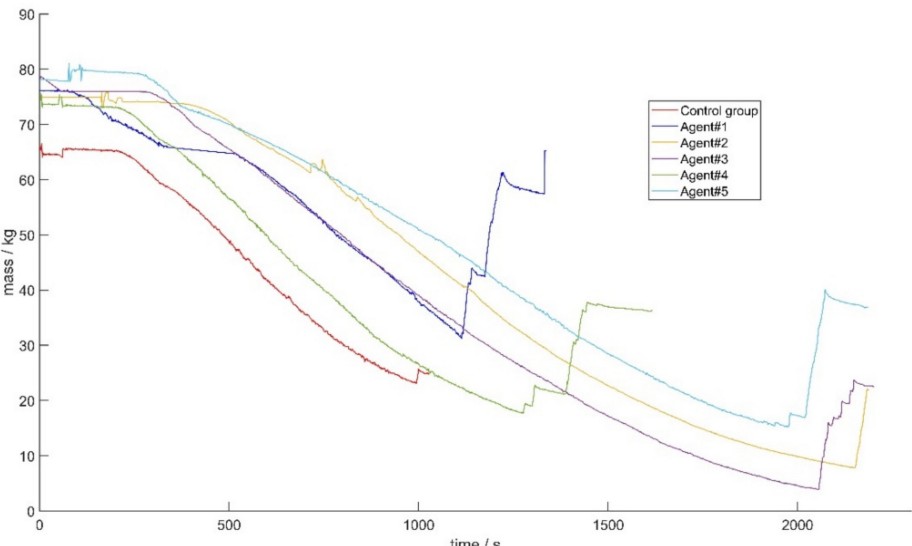

**Figure 25.** Differences in crib#3 mass losses after the helicopter had sprayed the flame retardants.

When the H125 helicopter flies at a height of 30 m and sprays 660 kg of flame-retardant water agent at a speed of 20 km/h, the following conclusions can be made after analyzing the combustion process and the above data:

(1) Compared with not spraying any flame-retardant water agent, spraying pure water, 10% Class AB flame retardant, 10% Class A flame retardant, or gel flame retardant can reduce the radiation intensity and mass loss due to the wood crib fire to a certain extent, thus prolonging the burning time of the wood crib fire;

(2) With regard to the mass loss index, the flame retardant properties from high to low are as follows: 10% Class A flame retardant + 0.6% guar gum > gel flame retardant > 10% Class A flame retardant > Class AB flame retardant > pure water. In terms of the radiant heat intensity index, the flame retardant properties from high to low are as follows: gel flame retardant ≈ 10% Class A flame retardant + 0.6% guar gum > 10% Class A flame retardant > Class AB flame retardant > pure water. Although the use of a flame-retardant water agent has little effect on retarding and reducing the temperature inside the combustibles, the temperature above the combustibles decreased significantly due to the reduction in fire intensity;

(3) When a wildfire spreads and expands, the water on the surface and inside of the surrounding combustibles will evaporate when the adjacent combustibles burn, thus reducing the effect of flame retardants, especially when the helicopter cannot completely cover the combustible surface. These uncovered surfaces are the weak points that lead to the spread of fires, and in this case, it is often impossible to stop the wildfire by spraying flame-retardant water agents;

(4) The temperature analysis of the thermocouples at 1 m, 1.7 m, and 2.7 m height shows that the temperature at the three heights can be effectively reduced, but the temperature inside the wood crib is not reduced.

In addition, the following fire prevention measure is also proposed: since the flame retardancy of Crib#1 is obviously higher than that of Crib#2, regular removal of any surface vegetation and the humus layer will reduce the thickness of the surface combustibles and enhance flame retardancy.

The mechanisms utilized by the different flame retardants are as follows:

(1) Pure water can only moisten the wood crib. The pure water in and on the wooden crib does not have a flame retardant role when it is completely evaporated by the standing and ignition processes. Water volume limitations and the short absorption time mean that the wood crib cannot be completely covered, resulting in a poor flame retardant effect;

(2) Class AB flame retardant has poor wind resistance, which means that its diffusion area is wider than those of Agent#1, Agent#3, and Agent#4. Furthermore, the amount of flame-retardant water agent per unit area is also less. When dropped from the air, the foam covers a wide area and helps limit the spread of the fire. Once dispersed on a fire, the foam absorbs heat from combustion while the bubble structure slowly releases water, which is absorbed by wood fuels. Foam improves the effectiveness of water by (1) helping water soak deeper and more quickly into forest fuels, such as wood, brush, and wood debris; and (2) slowing the evaporation of water held within the foam;

(3) The gel flame-retardant water agent has good water absorption performance and poor fluidity, which means that that the water utilization rate is high. The principle consists of two components: One is that super absorbent particles absorb water (hundreds of times their own weight) in a chemical-physical process called hydration. The stacked and water-filled "bubbles" greatly enhance the thermal protection performance of the flame-retardant water agent. The second one is to prevent the flame-retardant water agent from turning into steam in the superheated air above the wood crib fire and being taken away by the high-temperature smoke plume gas. The adhesive properties of the gel-based Agent#3 slow down the evaporation process, enabling more product to reach the fire source through hot air [32]. These two aspects need to be taken into account when gel flame retardant is sprayed from a helicopter. When the amount of flame-retardant water agent is less than that required for fire retardant (that is, the fire field intensity is large, for example, with the 12-layer wood crib when it is still fully burning after the flame retardant has been applied), the fire retardant performance is poor because the gel-based Agent#3 cannot cover all the burning points. When the amount of flame-retardant water agent is sufficient relative to the fire retardant requirements (that is, the intensity of the fire field is small; for example, a 4-layer wood cribs cannot be ignited after the flame retardant has been applied), a colloid can form on the surface of the wood strip that wraps around it to prevent the combustibles from being ignited. If there is not complete coverage (100%) on all surfaces, then gels are useless as the exposed area can catch fire and burn right through a structure. Therefore, helicopters should spray more gel flame retardants to cover the surface of combustibles as much as possible. A ground coating should be used as far as possible to protect wooden structures and improve helicopter spraying efficiency at reducing the intensity of the fire field;

(4) Class A flame retardant contains large amount of salts. When $(NH_4)_2CO_3$ decomposes, Class A flame retardant absorbs 48 kJ/mol more heat than pure water and evaporates water to rapidly cool down. It also generates inert gases, such as $NH_3$ and $CO_2$, to isolate oxygen. The phosphoric acid, metaphosphoric acid, and polymetaphosphoric acid produced in the chemical reactions can react with carbonaceous compounds and generate a dense and flame retardant coating over the surface of combustible materials, which can effectively delay the re-ignition time and reduce the fire intensity after re-ignition [33];

(5) Class A flame retardant + guar gum is a mixture of Agent#2 and Agent#3 and has both chemical flame retardant and physical flame retardant effects. It is consistent with the flame retardant principle outlined by Ref. [12] and its flame retardancy is relatively better than single flame retardants.

## 5. Applications

From August to September 2022, Hunan Province suffered the strongest drought since 1961 and industrial and residential electricity demand was extremely high. The load in Hunan, Jiangsu, and other provinces increased by more than 18% compared to the same period in 2021. Therefore, there was a great contradiction between electricity consumption and power supply. At the same time, the dry weather and low moisture content of combustibles meant it was extremely easy to ignite them and the risk of wildfires

in dense channels was high. In Shimen County, Changde, and Hunan, China, the crossing lines include the ±800 kV Fufeng Line #1494–1495, ±800 kV Jinsu Line #2104–2105, and ±800 kV Qishao Line #3964–3965, which are typical dense transmission lines. In order to ensure the efficient transmission of electricity during the summer peak season, on 22 August, the transmission load of the Fufeng Line was 4.76 million kW, that of the Jinsu Line was 6.83 million kW, and that of the Qishao Line was 6 million kW. A fire in the dense channel could very easily have caused multiple UHV transmission lines to trip at the same time, with considerable risk to the power grid. Since 23 August 2022, one H125 fire fighting helicopter has been deployed in central Hunan after reports of frequent wildfires.

On the afternoon of 25 August 2022, a forest fire broke out in Changde, as shown in Figure 26, which seriously threatened the operation of UHV lines. The vegetation on the site, which was mainly Chinese pine with a high oil content and an extremely high heat of combustion, was dense. At that time, the temperature was as high as 40 °C and the humidity was only 20%, which meant that the vegetation had a low moisture content. The fire spread rapidly up the mountain and an extreme fire tornado formed under agitation by the wind, which was incredibly dangerous. The burned area reached 220 hectares and the open fire area reached 45 hectares. The fire site was steep and inaccessible to ground personnel and there were no natural or artificial isolation zones.

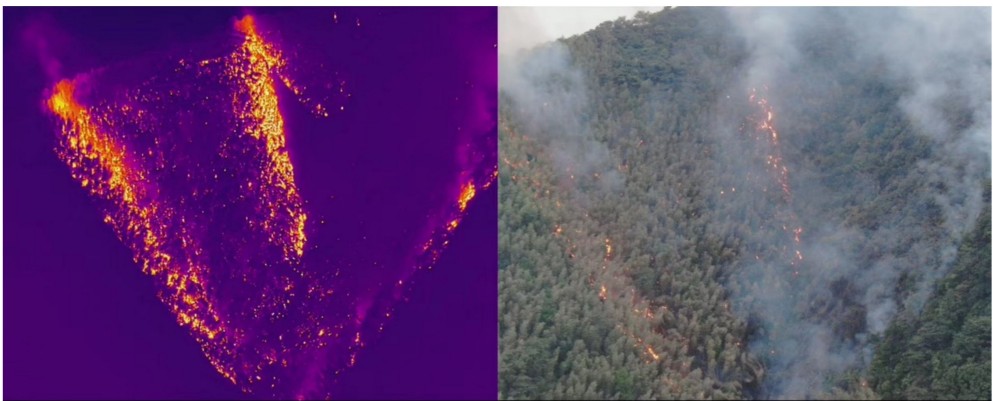

**Figure 26.** Monitoring map for the local location of on-site wildfires. Left image shows an infrared image of wildfire disasters and the right hand picture shows the visible light image of a wildfire disaster.

From the 26th to the 27th, the combined on-site terrain and meteorological conditions at 18:40 before sunset led to a helicopter being used to spray Class A flame retardant + guar gum when a downhill fire began in the canyon, as shown in Figures 27 and 28. A total of 85 barrels of flame-retardant water agent, 68 tons in total, were sprayed for 4.5 h to suppress the fire site at the smoke point. When the temperature decreased and the humidity increased at night, ground personnel used the No. 2 tool and a multi-stage water pump to relay a water supply to the fire site to clean up the smoke point at a fixed point. This completely extinguished the fire, avoided the burning of houses and a large amount of economic vegetation, ensured the safety of the lives and property of the local people, and effectively reduced the threat of wildfires near sub-transmission lines that were associated with important transmission lines.

During the summer peak season of 2022, the Hunan Power Grid dispatched a helicopter 64 times, sprinkled 844 buckets of water, and the cumulative firefighting time was 83.5 h. They successfully extinguished or timely blocked 41 wildfires on transmission lines, such as the 500 kV Changmin and Jinhong Lines and the 220 kV Tizhong Line, as shown in Table 5. There were no 220 kV wildfire tripping accidents on the transmission lines with voltage levels of 220 kV or above.

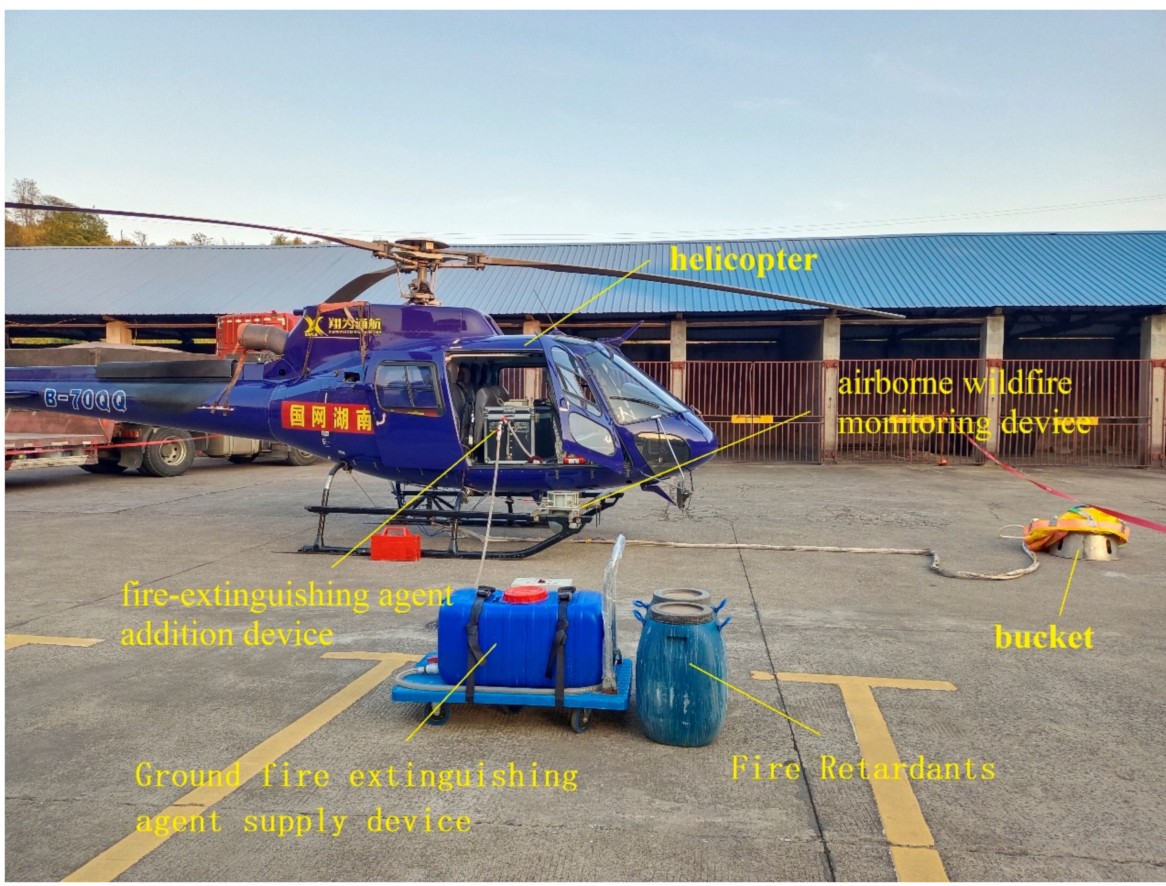

**Figure 27.** Adding a flame retardant to a firefighting helicopter.

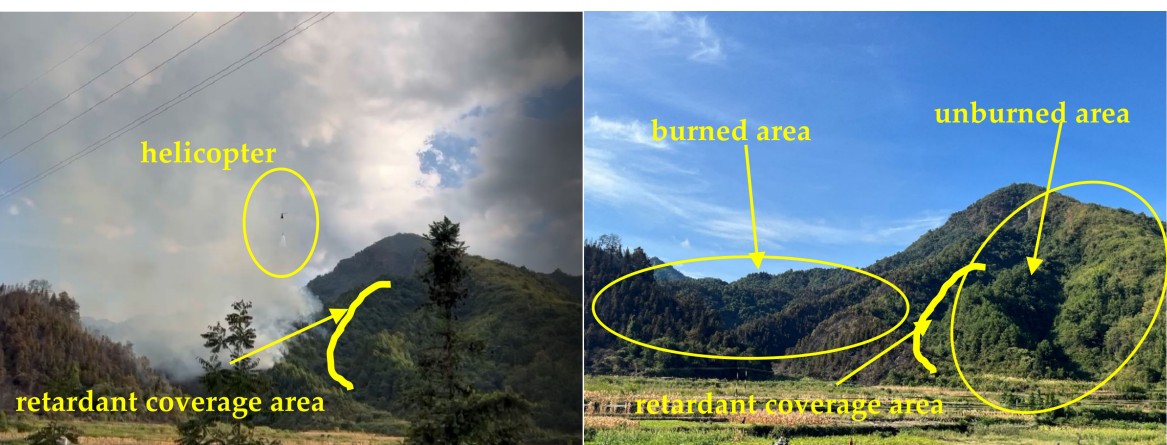

**Figure 28.** An example of using a helicopter to a spray flame-retardant water agent to quickly extinguish a large-scale wildfire on site. The picture on the left shows that the flame retardant is sprayed to form an isolation belt and the picture on the right shows that the wildfire was quickly extinguished at the spraying position without further spread.

**Table 5.** Fire extinguishing and the application of flame retardants to forest fires in the Hunan Power Grid area during the Summer peak season in 2022.

| Location | Date | Flight Time | Number of Buckets | Amount of Flame Retardant (kg) | Fire Extinguishing Line |
|---|---|---|---|---|---|
| Changde | 26–27 August | 13 h 51 min | 93 | 2258 | Fire in Shimen County |
| Loudi | 4 September | 59 min | 5 | 80 | 500 kV Jinhong Line #003 |
| Loudi | 5 September | 2 h 21 min | 18 | 80 | 500 kV Changmin Line #179–#180, 220 kV Qunkang Line |
| Loudi | 6 September | 2 h 44 min | 27 | 50 | 500 kV Changmin Line #179–#181 |
| Loudi | 7 September | 4 h 33 min | 53 | 100 | 500 kV Changmin Line #044–#046 |
| Loudi | 8 September | 8 h 56 min | 97 | 150 | Forest fire in Weishan Township, Xinhua County, Loudi City |
| Loudi | 1 September | 1 h 6 min | 8 | 30 | 500 kV Changmin Line #192 |
| Loudi | 11 September | 3 h 34 min | 50 | 80 | 220 kV Tizhong Line #27 |
| Yongzhou | 14 September | 5 h 18 min | 16 | 80 | Forest fire in Huangjiangyuan Village, Yongzhou |
| Yongzhou | 15–17 September | 18 h 45 min | 92 | 450 | |
| Loudi | 19 September | 1 h 41 min | 13 | 50 | 500 kV Hongmin Line 2#74–#75 |
| Loudi | 23 September | 49 min | 5 | 50 | 500 kV Jinhong Line I #003, 220 kv Jinti Line III #5 |
| Loudi | 24 September | 3 h 32 min | 30 | 100 | 220 kV Hongbao Line #59 |
| Loudi | 26 September | 2 h 2 min | 13 | 50 | 220 kV Tiqun Line I #78–79 |
| Loudi | 27 September | 1 h 35 min | 8 | 40 | 220 kV Liankang Line 1#7–#8 |
| Loudi | 30 September | 4 h 4 min | 37 | 100 | 220 kV Tihe Line I #13–#14 |

## 6. Conclusions

In this study, an H125 helicopter was used to carry out flame retardant tests with 660 kg of different flame-retardant agents. The flame retardant effect of the different flame-retardant water agents on the wood cribs was observed with conclusions drawn as follows:

(1) Compared to not spraying any flame-retardant water agent, pure water, Class AB flame retardants, Class A flame retardants, gel flame retardants, etc., can reduce the intensity of a wood crib fire to a certain extent;

(2) The mass loss index results showed that the flame retardancy from high to low was 10% Class A flame retardant + 0.6% guar gum > gel flame retardant > 10% Class A flame retardant > Class AB flame retardant > pure water. The radiant heat intensity index results showed that flame retardancy from high to low was gel flame retardant ≈ 10% Class A flame retardant + 0.6% guar gum > 10% Class A flame retardant > Class AB flame retardant > pure water;

(3) Based on the flame-retardant properties of the different flame retardants tested in this study, they were applied in Loudi, Changde, and other cities in Hunan Province to prevent and control wildfire disasters near transmission lines. They effectively ensured the safe operation of dense power grid channels during the high-incidence period for wildfires in the power grid area and under extreme dry weather conditions;

(4) In this study, only laboratory experiments were carried out; the impact of ladder combustible on the flame-retardant properties has not been studied. This method can be studied more thoroughly in future wildfire experiments.

**Author Contributions:** Conceived and designed the paper: J.L. Data collection: T.Z. Testing and data analysis: T.Z. and Y.O. Wrote the paper: T.Z. and C.W. All authors have read and agreed to the published version of the manuscript.

**Funding:** This work was supported by the State Grid Major S&T Project (No. 5216A0210041).

**Institutional Review Board Statement:** Not applicable.

**Informed Consent Statement:** Not applicable.

**Data Availability Statement:** The data presented in this study are available on request from the corresponding author.

**Acknowledgments:** The authors would like to thank the Editor and the reviewers whose comments and suggestions have been very helpful in improving the quality of this paper.

**Conflicts of Interest:** The authors declared that they have no conflict of interest to this work.

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
