# Peer review of "Dropping Fire Retardants by Helicopter and Its Application to Wildfire Prevention near Electrical Transmission Lines"

_fire, doi:10.3390/fire6050176_

Round 1

Reviewer 1 Report

It is recommended to improve the content from report format to thesis format drastically. Figures and tables should be accurately linked to the contents of the text, and try to improve the reader's understanding by avoiding long sentence explanations in the text.

Please reinforce a clear description of experimental conditions and methods throughout and also try to improve your understanding rather than a quick reply. Should frame the manuscript well concerning the position and size of figures/tables.

Graphs should be written with titles and descriptions separated. (Explanation should be included in the text, not in the title.)

 The following should be reflected in the manuscript.

 1: The article must be amended to include up to two reference numbers when creating reference numbers. This measure conveys specific information to the reader precisely. References 3 to 6, 8 to 12 and 17 to 20 correspond to these.

2. It is essential for the author to include a flowchart for linking research from parts 1 to 5 to increase readers' understanding, so adding a flowchart is recommended.

3. Please, clarify the meaning of 'HeatFlowSen' of serial no.17 in table1.

4. Authors should add photos or drawings of the geometry of the wood cribs applied to the 2.1 test combustible material.

5. In 2.2 Fire Scene Layout and Measurement Scheme, the distribution of ground flame retardants released by helicopters must be scribed numerically. This will help improve the quality of this study since the fire extinguishing properties vary depending on the amount and distribution of the fire extinguishing agent sprayed.

6. The position of the thermocouple must be reflected in the drawing.

7. In Fig. 1, the authors applied different values: the distance between Crib#1 and Crib#2 is 8.4m, and between Crib#2 and Crib#3 is 3.6m. What is the reason for this?

8. The purpose for which the electronic balance had been installed under Array#1, Array#2, and Array#3 must be explained clearly.

9. In this research, three cameras were used during the test to determine whether the area covered by the flame-retardant water agent completely covered the three wood cribs in the flight direction. These results should be combined in 3.2 'Tests on Flame-retardant Water Agents when Sprayed from a Helicopter' with photographs obtained by cameras. This will significantly contribute to enhancing the authenticity of this research.

10. In Figure 5, the title and discussion of the figure should be written separately.

11. Wood cribs are sensitive to humidity, so they must be prepared under the same experimental conditions in which they are used. Preparatory conditions for this should be described.

12. The reason for applying Guar gum to Flame Retardant Water Agents should be explained. As an applied additive, is there a problem with guar gum separation from the aqueous solution? Why was it selected at 0.6wt%?

13. In Figure 3 for the Diagram of the crib apparatus, shouldn't one oil pan be applied to each crib?

14. If one oil pan was applied to two cribs, there is a possibility of developing uneven combustion.

15. According to Thermocouple No., the specific location of the Thermocouple must be indicated in your figure.

16. To help the reader's understanding, it is recommended to process and provide the experimental data with maximum readability. Figure 5~11 corresponds to this. Organizing the data and showing the difference by the condition in a neat comparison graph is recommended.

17. The composition of the Class A & AB Flame Retardant should be mentioned.

18. It is necessary to reinforce the specificity of the correlation between laboratory and wildfire experiments in your result.

end/

Author Response

We sincerely thank you for your valuable time and comments on our manuscript. We have taken your comments very seriously and have tried our best to revise the manuscript accordingly. Should you have any other questions, please let us know.

Reviewer 2 Report

The authors numerically studied the effect of five fire retardants dropped from helicopter on the burning intensity of the wood cribs fire. The manuscript is well-organized, and the results are clearly presented. The topic of the paper is of interest to the fire community. The results are valuable for wildland fire suppression. I think this paper deserves to be published in fire. While the authors should consider the following questions and comments as they revise the manuscript.

1. In the analysis of the flame retardant mechanism in the article, it is mentioned that the viscosity of the fire extinguishing agent affects the flame retardancy. Please provide the viscosity of the three flame retardant (0.3% gel flame retardant, 10% Class A flame retardant, and 10% Class A flame retardant + 0.6% guar gum).

2. There are no references to "Figure 12", "Figure 13", "Figure 14" in the manuscript.

Author Response

Sincerely thank you for your review and recommendation. Your recommendation has encouraged us to continue to improve the manuscript. We have given serious thoughts to your comments and tried our best to make the corresponding modification. If there is anything not clear, please do not hesitate to tell us again.

Round 2

Reviewer 1 Report

Finally, the article recommends using the MDPI English translation service